# The impact of antimalarial resistance on the genetic structure of *Plasmodium falciparum* in the DRC

Robert Verity [1,17✉], Ozkan Aydemir [2,17], Nicholas F. Brazeau [3,17], Oliver J. Watson [1], Nicholas J. Hathaway[4], Melchior Kashamuka Mwandagalirwa[5], Patrick W. Marsh[2], Kyaw Thwai[3], Travis Fulton[6], Madeline Denton[6], Andrew P. Morgan [6], Jonathan B. Parr[6], Patrick K. Tumwebaze[7], Melissa Conrad[8], Philip J. Rosenthal[8], Deus S. Ishengoma[9], Jeremiah Ngondi [10], Julie Gutman [11], Modest Mulenga[12], Douglas E. Norris[13], William J. Moss[14], Benedicta A. Mensah[15], James L. Myers-Hansen [15], Anita Ghansah[15], Antoinette K. Tshefu[5], Azra C. Ghani[1], Steven R. Meshnick[3], Jeffrey A. Bailey [2,18] & Jonathan J. Juliano [3,6,16,18✉]

The Democratic Republic of the Congo (DRC) harbors 11% of global malaria cases, yet little is known about the spatial and genetic structure of the parasite population in that country. We sequence 2537 *Plasmodium falciparum* infections, including a nationally representative population sample from DRC and samples from surrounding countries, using molecular inversion probes - a high-throughput genotyping tool. We identify an east-west divide in haplotypes known to confer resistance to chloroquine and sulfadoxine-pyrimethamine. Furthermore, we identify highly related parasites over large geographic distances, indicative of gene flow and migration. Our results are consistent with a background of isolation by distance combined with the effects of selection for antimalarial drug resistance. This study provides a high-resolution view of parasite genetic structure across a large country in Africa and provides a baseline to study how implementation programs may impact parasite populations.

[1] Medical Research Council Centre for Global Infectious Disease Analysis, Department of Infectious Disease Epidemiology, Imperial College London, London, UK. [2] Department of Pathology and Laboratory Medicine, Warren Alpert Medical School, Brown University, Providence, RI, USA. [3] Department of Epidemiology, Gillings School of Global Public Health, University of North Carolina, Chapel Hill, USA. [4] Program in Bioinformatics and Integrative Biology, University of Massachusetts, Worcester, MA, USA. [5] Kinshasa School of Public Health, Hôpital Général Provincial de Référence de Kinshasa, Kinshasa, Democratic, Republic of Congo. [6] Division of Infectious Diseases, Department of Medicine, School of Medicine, University of North Carolina at Chapel Hill, Chapel Hill, NC, USA. [7] Infectious Disease Research Collaboration, Kampala, Uganda. [8] Department of Medicine, University of California- San Francisco, San Francisco, CA, USA. [9] National Institute for Medical Research, Tanga, Tanzania. [10] RTI International, Dar es Salaam, Tanzania. [11] Malaria Branch, Center for Global Health, Centers for Disease Control, Atlanta, GA, USA. [12] Tropical Disease Research Centre, Ndola, Zambia. [13] Department of Molecular Microbiology and Immunology, Johns Hopkins Bloomberg School of Public Health, Baltimore, MD, USA. [14] Department of Epidemiology, Johns Hopkins Bloomberg School of Public Health, Baltimore, MD, USA. [15] Noguchi Memorial Institute for Medical Research, University of Ghana, Accra, Ghana. [16] Curriculum in Genetics and Molecular Biology, School of Medicine, University of North Carolina at Chapel Hill, Chapel Hill, NC, USA. [17] These authors contributed equally: Robert Verity, Ozkan Aydemir, Nicholas F. Brazeau. [18] These authors jointly supervised this work: Jeffrey A. Bailey, Jonathan J. Juliano. ✉email: r.verity@imperial.ac.uk; jjuliano@med.unc.edu

Malaria remains one of the largest global public health challenges, with an estimated 219 million cases worldwide in 2017[1]. Despite decades of scale-up in control, there has been a recent resurgence, particularly in high transmission countries in sub-Saharan Africa[1]. In addition, the emergence of antimalarial resistance poses a major threat to current control and elimination efforts worldwide, and new tools are needed to quantify the changing landscape of drug resistance on timescales relevant to malaria control programmes. Genomics has emerged as an useful method for better understanding parasite populations that can be leveraged to support the design of effective interventions against a continually evolving parasite.

Data from genomic studies provides information that is complementary to epidemiological data[2], and can help to answer several key questions, including how parasites are transmitted, how drug resistance spreads, and how malaria control efforts impact the diversity of the parasite population. However, to date, efforts to use genomics to inform malaria control efforts have suffered from three major limitations. First, much of the work has been conducted in low transmission regions, such as Asia and transmission fringe regions of Africa, leaving it unclear how useful information can be gathered in the highest transmission settings. Some of these high burden regions have experienced increasing malaria prevalence in recent years and are now the center of strategic plans for control efforts[3,4]. Second, most genomic studies in Africa have relied upon convenience sampling from a few sites usually collected for other purposes, rather than population-representative samples. Lastly, studies have either relied on relatively few genetic markers, providing limited insight into the complete genome, or on expensive whole-genome sequencing, limiting the number of samples studied. Overcoming these limitations is essential for genomics to have broader impacts on malaria control.

Within Africa, parasite populations have been shown to vary significantly between East and West, as demonstrated by their distinct antimalarial drug susceptibilities and population genetics[5,6]. However, few genomic studies have incorporated samples from central Africa, limiting our understanding of the connectivity of parasite populations across the continent. The Democratic Republic of the Congo (DRC) is the largest malaria-endemic country in Africa, borders nine countries, and harbors ~11% of global *P. falciparum* malaria cases[1]. The DRC harbors a large, understudied parasite population that likely serves as a bridge between African parasite populations. Limited previous work has shown that the DRC represents a watershed between East and West African drug resistant parasite populations for sulfadoxine-pyrimethamine and chloroquine resistance[7–9]. More recently, parasite population structuring due to mutations at these and other loci associated with antimalarial resistance has been confirmed within the DRC[10]. However, studies focusing on hypervariable surface antigen diversity or neutral microsatellites have been unable to detect significant structure in the parasite population[10,11], likely due to a lack of high-quality genome-wide signal. A better understanding of parasite populations and the spread of antimalarial resistance in the DRC will allow for the design of more effective interventions accounting for evolutionary forces.

To address this knowledge gap, we leverage a recent advance in malaria genomics, high-throughput molecular inversion probe (MIP) capture and sequencing, to characterize and map parasite population structure and antimalarial resistance profiles in the DRC and to define the connections of parasites within the DRC to East and West African parasite populations[12]. This approach provides a cost-effective and scalable method of genome interrogation, without the expense or informatic complexities of whole-genome sequencing. We previously employed MIPs to comprehensively genotype known antimalarial resistance genes in several hundred samples from the DRC[10]. Here, we introduce an expanded MIP panel targeted at 1834 single-nucleotide polymorphisms (SNPs) distributed throughout the *P. falciparum* genome, and designed to quantify differentiation and relatedness between samples. Using this panel of genome-wide SNP MIPs, in combination with the previous drug resistance MIP panel, we evaluate the parasite population diversity in 2537 parasite isolates from the DRC and surrounding countries in East and West Africa. We use this information to quantify relatedness of and gene-flow between parasites over large geographic scales and to assess the origins of antimalarial resistance mutations.

## Results

**Sample quality and filtering**. We obtained 2537 samples collected in 2013–2015 from the DRC and surrounding countries (DRC = 2039, Ghana = 194, Tanzania = 120, Uganda = 63, Zambia = 121). All samples were sequenced using two separate MIP panels: a genome-wide panel designed to capture overall levels of differentiation and relatedness, and a drug resistance panel designed to target polymorphic sites known to be associated with antimalarial resistance[10]. The genome-wide panel included 739 ostensibly geographically informative SNPs, chosen on the basis of high differentiation ($F_{ST}$) between surrounding African countries in publicly available genomic sequences made available by the Pf3K project (see Supplementary Note 1 and Supplementary Data 1), and 1151 putatively neutral SNPs distributed throughout the genome, with an overlap of 56 SNPs that were both neutral and geographically informative. The drug resistance panel included SNPs in known and putative drug resistance genes and has been described elsewhere[10]. The median number of unique molecular identifiers (UMIs) per MIP was 31 (range: 1–8490) for the genome-wide panel, and 10 (range: 1–32,511) for the drug resistance panel. Complete UMI depth distributions are shown in Supplementary Fig. 1. After filtering for samples and loci with sufficient UMI coverage, we were left with 1382 samples and 1079 loci from the genome-wide panel, and 674 samples and 1000 loci from the drug resistance panel, with an overlap of 452 samples between both panels. In addition to these samples, 114 controls consisting of known mixtures were sequenced and used to assess the accuracy of allele calls and frequencies. Expected versus measured allele frequencies for each SNP, calculated from these controls, are shown in Supplementary Fig. 2.

**Complexity of infection in the DRC**. Initial analyses focused on the genome-wide MIP panel only. Complexity of infection (COI) for each sample was estimated using THE REAL McCOIL[13] (Supplementary Fig. 3). The mean COI was estimated at 2.2 (range 1–8) for the study as a whole. We observed significant differences in COI between countries (Ghana: 1.55 (non-parametric bootstrap 95% CI: 1.39–1.73), DRC: 2.23 (2.15– 2.31), Tanzania: 2.17 (1.83–2.51), Zambia: 2.68 (2.39–3.00), Uganda 2.18 (1.87– 2.51), and within the DRC we observed a statistically significant relationship between COI and *P. falciparum* prevalence by microscopy at both the province and cluster levels (Supplementary Fig. 4), with higher COIs observed at higher prevalences.

**Population structure in the DRC**. We explored population structure through principal component analysis (PCA) evaluated on within-sample allele frequencies at all 1079 genome-wide loci. We found the same separation between East and West Africa described in previous studies (Fig. 1) as well as finer structure between regions within East Africa. DRC samples comprised a continuum between the East and West African clusters.

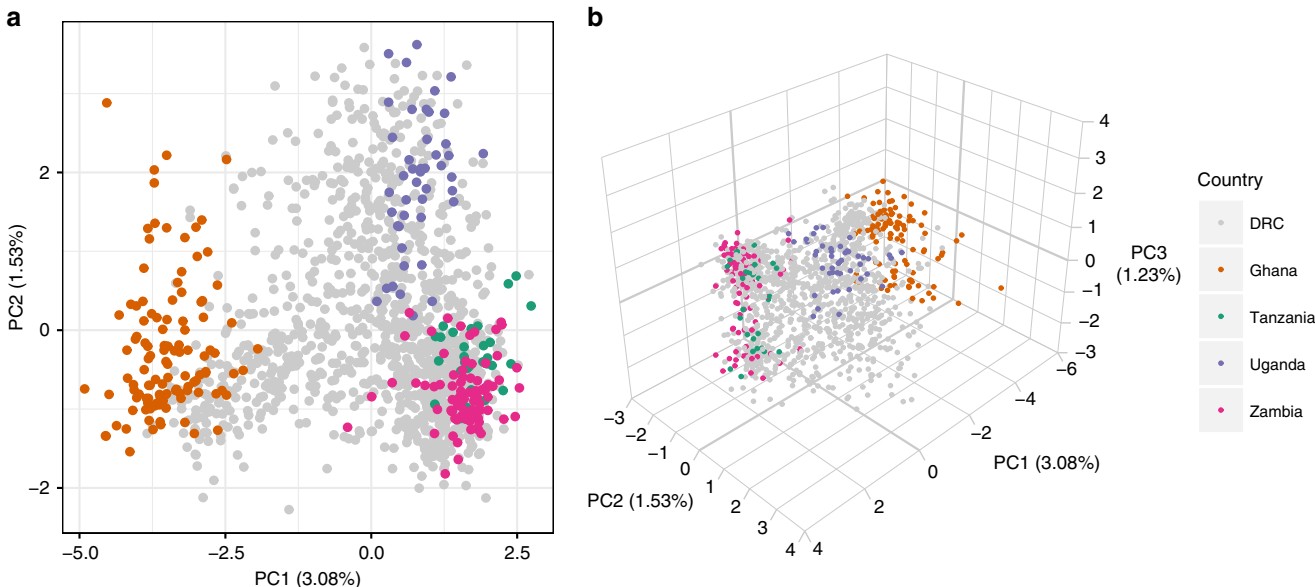

**Fig. 1 Principal component analysis.** The first two (**a**) and three (**b**) principal components calculated from within-sample allele frequencies using the genome-wide MIP panel. Colors indicate country of origin of each sample.

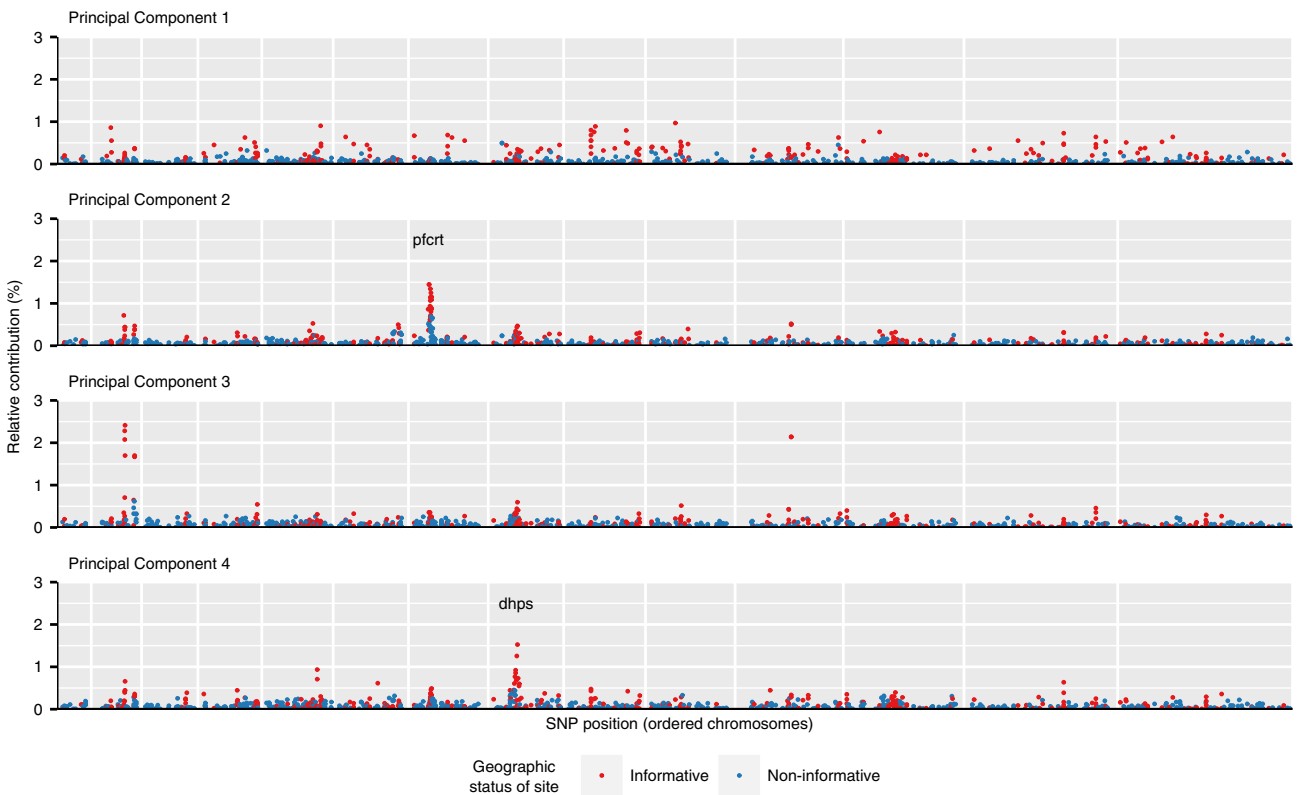

**Fig. 2 Per-locus contributions to principal components.** The relative contribution (%) of each locus to the first four principal components. Chromosomes are plotted in order, separated by vertical white gridlines. Point colors indicate sites that were chosen in the design based on $F_{ST}$ values to be geographically informative (blue) or not (red).

The relative contribution of each locus to each principal component was quantified through normalized loading values. Relative contributions to the first four principal components are shown in Fig. 2. After the fourth principal component the percent variance explained by subsequent components plateaued (Supplementary Fig. 5). For principal component 1 (PC1) large contributions came from loci distributed throughout the genome,

and a relatively larger contribution (65.2%) came from putatively geographically informative SNPs (non-parametric bootstrap, $p <$ 0.001). In contrast, contributions to PC2 were concentrated in a region on chromosome seven in close proximity to *P. falciparum* chloroquine resistance transporter (*pfcrt*), a known drug resistance locus, suggesting that resistance to chloroquine or amodiaquine may be driving differentiation along this secondary

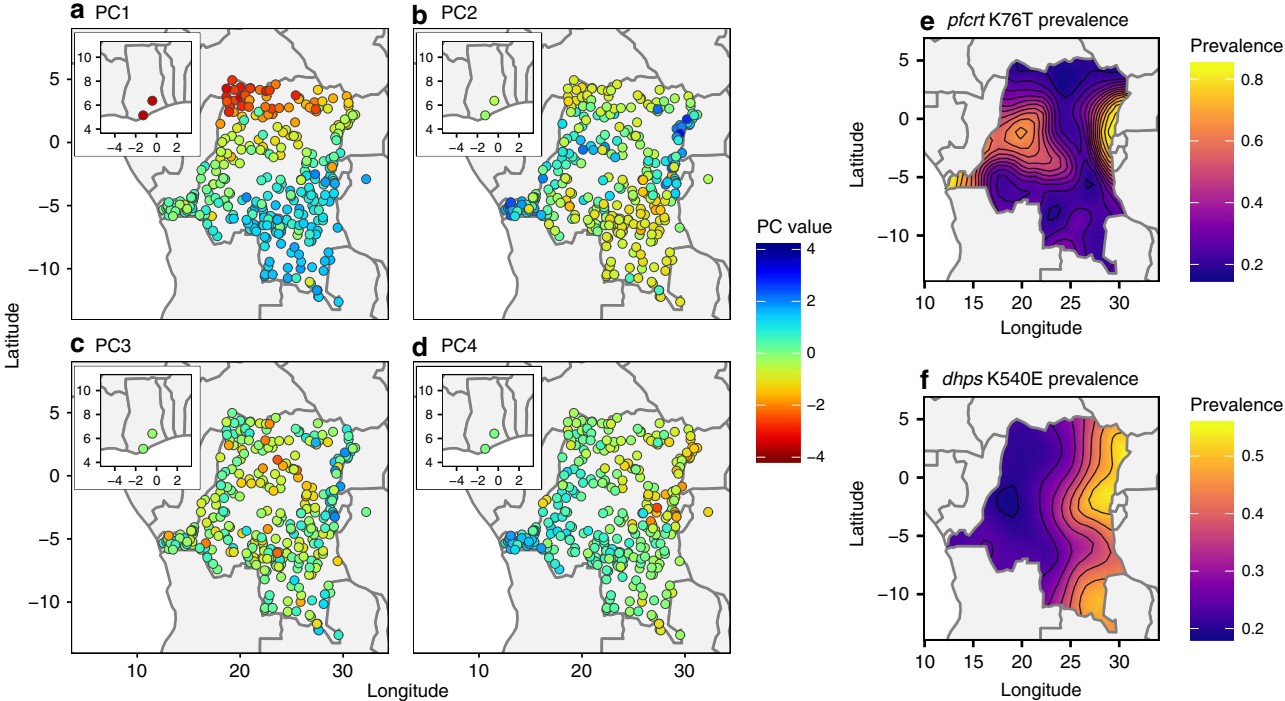

**Fig. 3 Spatial patterns in principal components. a–d** Show the mean principal component value per DHS cluster. **e, f** Show estimated distributions of the prevalence of molecular markers of resistance for *pfcrt* and *pfdhps*.

axis. For PC3, locus contributions were concentrated in three genic regions: PF3D7_0215300 (8.5%), PF3D7_0220300 (5.0%), and PF3D7_1127000 (4.3%). The first and largest of these encodes an acyl-CoA synthetase and is part of a diverse gene family known to undergo extensive gene conversion and recombination[14]. For PC4 we observed a region of high locus contribution on chromosome eight in close proximity to the known antifolate drug resistance gene dihydropteroate synthase (*dhps*). Combined, these results suggest that geography and drug resistance are both contributors to the observed population structure.

The relationship between the PCA results and the spatial distribution of parasites was explored by plotting raw principal component values against the geographic location of samples (Fig. 3a–d). For PC1 this revealed a complex pattern of spatial variation, containing both north–south and east–west clines. For PC2 and PC4 the maps essentially recapitulate the known geographic distribution of *pfcrt* and *dhps* resistance mutations, respectively (Fig. 3e, f). For PC3 the map indicates some east–west spatial structuring that is not explained by known markers of antimalarial resistance and warrants further investigation.

**Between sample relatedness of parasites**. The relatedness of all pairs of samples was explored through pairwise identity by descent (IBD), estimated using a maximum likelihood approach. IBD describes the relatedness of samples in terms of their shared evolutionary history, and consequently is not influenced by a particular allele frequency distribution. This makes it a better measure than simple identity by state (IBS) when comparing between studies, as values can be compared directly[15]. We first carried out a simulation-based analysis to explore the accuracy of our maximum likelihood estimator (see Supplementary Note 2 and Supplementary Fig. 6), finding that we were conservatively biased in cases of high polyclonality. Hence, we expect to underestimate true IBD by this method. This result did depend

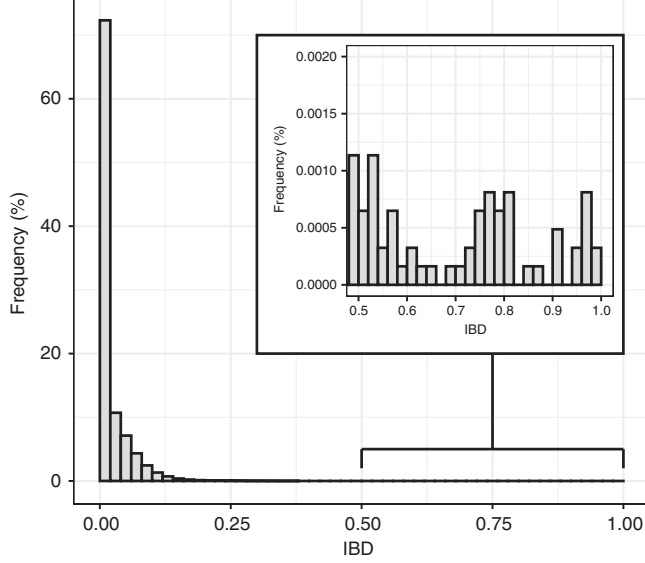

**Fig. 4 Histogram of pairwise IBD.** Pairwise IBD between all samples, estimated by maximum likelihood. Inset shows the heavy tail of the distribution, with some pairs of samples having IBD > 0.9.

on the number of genotyped positions, with estimates becoming increasingly unreliable for smaller datasets of 100 or 20 SNP loci. In the real data, the overall distribution of pairwise IBD was found to be heavy-tailed, consisting of a large body of weakly related samples and a tail of very highly related samples (Fig. 4).

Mean IBD was significantly higher within clusters compared to between clusters (0.06 vs. 0.02, two-sample *t*-test, $p < 0.001$). When plotted against geographic separation there was a clear fall-off of IBD with distance (Fig. 5a), consistent with the classical pattern expected under isolation-by-distance[16,17]. Focussing on the tail of highly related samples, which includes the major strain in complex infections, there were 12 sample pairs with a

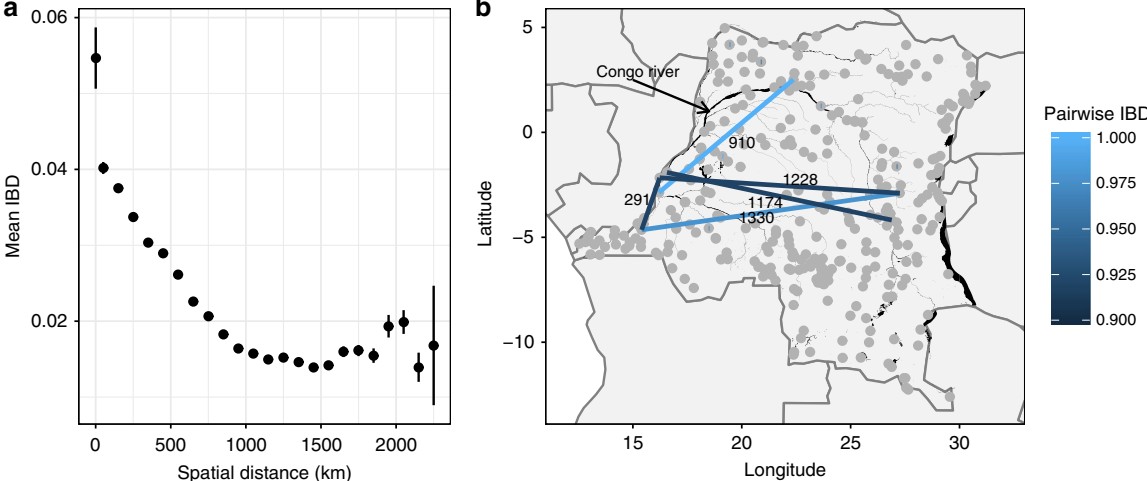

**Fig. 5 Spatial patterns in IBD. a** Shows the mean IBD between clusters, binned by the spatial distance between clusters. Vertical lines show 95% confidence intervals. **b** Shows the spatial distribution of highly related (IBD > 0.9) parasite pairs. Values above edges give distances in km. Black areas indicate major water bodies, including the Congo River, which is labeled.

relatedness greater than IBD = 0.9. Comparison of raw allele frequency distributions confirmed that these were likely clones (Supplementary Fig. 7). These highly related pairs were found more often within the same cluster than in different clusters (7 vs. 5, respectively, chi-squared test, $p < 0.001$), suggesting the presence of local clonal transmission chains. The five between-cluster highly related pairs (Fig. 5b) were spread over large geographic distances (281–1331 km), far beyond the normal expected scale of the breakdown in genetic relatedness (Fig. 5a), suggesting recent long distance migration.

**Prevalence of markers of antimalarial resistance.** Based on previous findings of an east–west divide in molecular markers of antimalarial resistance in the DRC[8,9], all samples in the DRC were divided by geographically weighted K-means clustering into two populations (Supplementary Fig. 8). The prevalence of every mutation identified by the drug resistance MIP panel was calculated in eastern and western DRC, as well as at the country level. Table 1 gives a summary of all mutations that reached a prevalence >5% in any geographic unit, and a complete list of all identified mutations along with their prevalence is given in Supplementary Data 2. Note that in the *dhps* mutation **G**437A the reference is resistant, hence this is re-coded as A437**G** and pre-valence values indicate the prevalence of the reference allele. Estimated prevalences of these alleles in the DRC as a whole were broadly similar to previously published estimates[10]. However, we did identify several polymorphisms in known and putative resistance genes not previously reported in the DRC, including *kelch* K189**T** and *pfatp6* N569**K**, both of which have been described at appreciable frequencies elsewhere in Africa[18–20].

**Geographic distribution of antimalarial resistant haplotypes.** Previous studies have demonstrated that mutations associated with antimalarial resistance are clustered into east–west group-ings within DRC[8,10]. Focusing on the 107 samples from DRC that were identified as monoclonal from the REAL McCOIL analysis, we explored the joint distribution of all combinations of mutant haplotypes in both the *dhps* and *crt* genes. Raw combinations of mutations were visualized using the UpSetR package in R[21], and the spatial distribution of haplotypes in the DRC was explored by plotting these same mutant combinations against their corre-sponding DHS cluster locations (Fig. 6). Our results for *dhps* recapitulate those found previously, showing a clear east–west

divide with the K540**E** and A581**G** mutants concentrated in the east, and S436**A** and **A**437**G** concentrated in the west. For *crt* we also find evidence of an east–west divide, with haplotypes con-taining N326**S** and F325**C** concentrated in the east and those containing I356**T** concentrated in the west.

**Selective sweep and haplotype analysis of antimalarial resis-tance.** Using the antimalarial resistance MIPs and genome-wide SNP MIPs combined, the extended haplotypes of the monoclonal infections were determined for 200 kb upstream and downstream of each putative drug resistance allele that had at least 5% overall prevalence in the DRC. The CV**IET** haplotype within the *crt* gene showed a signal of positive selection, with longer haplotype blocks in western DRC as compared to eastern DRC (Fig. 7; p'XP-EHH$_D$ < 0.05). In the east, patterns of haplotype homozygosity are consistent with positive selection for the derived I356**T** hap-lotype (Supplementary Fig. 9), although a XP-EHH$_D$ statistic could not be calculated for this locus because the derived hap-lotype was absent in western DRC, supporting the geographic localization of the I356**T** mutation in the east (Fig. 6).

Mutations in *dhps* were more difficult to interpret. This gene has undergone multiple selective sweeps associated with increas-ing drug resistance. The most recently introduced mutation into the DRC, *dhps* A581**G**, showed relatively conserved local haplotypes around the mutation in both eastern and western DRC (Supplementary Fig. 10). Extended haplotypes around the other mutations (Supplementary Figs. 11 and 12) are inconsistent with a classical hard sweep, perhaps due to selection on multiple independent haplotypes or to interference between A581**G** and other linked alleles. Finally, we did not detect any strong signals of differing patterns of recent positive selection between the eastern and western DRC among the *dhfr* and *mdr2* genes (Supplementary Table 1, Supplementary Fig. 13).

**Discussion**
Here we provide the first large-scale, robustly sampled study of *falciparum* malaria in central Africa using MIP capture and sequencing, a high-throughput genotyping approach that is appropriate for large population based surveys. Using a panel of probes designed to detect genome-wide SNPs, combined with a second panel targeting drug resistance genes, we were able to show that the parasite population in the DRC contains a signal of differentiation by geographic separation, consistent with the

**Table 1 Prevalence (%) of mutations identified by the drug resistance MIP panel.**

| | | | | Prevalence | | | | | | |
|---|---|---|---|---|---|---|---|---|---|---|
| Gene | Chromosome | Position | Mutation Name | Overall | DRC | DRC West | DRC East | Ghana | Uganda | Zambia |
| atp6 | chr1 | 267007 | I723V | 1.1 | 0.3 | 0.7 | 0.0 | 4.2 | 7.3 | 0.0 |
| atp6 | chr1 | 267257 | G639D | 2.0 | 1.8 | 2.9 | 1.0 | 0.0 | 7.3 | 0.0 |
| atp6 | chr1 | 267467 | N569K | 24.1 | 21.9 | 18.8 | 24.0 | 16.7 | 41.5 | 28.9 |
| atp6 | chr1 | 267882 | E431K | 15.3 | 17.0 | 18.8 | 15.7 | 16.7 | 9.8 | 6.7 |
| atp6 | chr1 | 267970 | L402V | 7.1 | 8.2 | 10.1 | 6.9 | 12.5 | 0.0 | 2.2 |
| dhfr-ts | chr4 | 748239 | N51I | 83.0 | 79.5 | 81.2 | 78.4 | 75.0 | 100.0 | 97.8 |
| dhfr-ts | chr4 | 748262 | C59R | 71.2 | 63.2 | 63.0 | 63.2 | 95.8 | 95.1 | 97.8 |
| dhfr-ts | chr4 | 748410 | S108N | 97.8 | 97.1 | 97.1 | 97.1 | 100.0 | 100.0 | 100.0 |
| dhfr-ts | chr4 | 748577 | I164L | 3.1 | 0.6 | 0.0 | 1.0 | 0.0 | 29.3 | 0.0 |
| mdr1 | chr5 | 958145 | N86Y | 12.4 | 14.3 | 18.8 | 11.3 | 16.7 | 7.3 | 0.0 |
| mdr1 | chr5 | 958440 | Y184F | 37.4 | 36.5 | 39.9 | 34.3 | 58.3 | 31.7 | 37.8 |
| mdr1 | chr5 | 958484 | T199S | 1.3 | 0.0 | 0.0 | 0.0 | 0.0 | 14.6 | 0.0 |
| mdr1 | chr5 | 958584 | S232Y | 2.7 | 3.5 | 5.1 | 2.5 | 0.0 | 0.0 | 0.0 |
| mdr1 | chr5 | 961625 | D1246Y | 4.4 | 2.9 | 3.6 | 2.5 | 0.0 | 24.4 | 0.0 |
| crt | chr7 | 403620 | M74I | 30.3 | 28.7 | 37.7 | 22.5 | 16.7 | 85.4 | 0.0 |
| crt | chr7 | 403621 | N75E | 30.3 | 28.7 | 37.7 | 22.5 | 16.7 | 85.4 | 0.0 |
| crt | chr7 | 403625 | K76T | 30.3 | 28.7 | 37.7 | 22.5 | 16.7 | 85.4 | 0.0 |
| crt | chr7 | 404407 | A220S | 28.1 | 24.6 | 31.9 | 19.6 | 8.3 | 100.0 | 0.0 |
| crt | chr7 | 405600 | I356T | 7.1 | 9.4 | 21.0 | 1.5 | 0.0 | 0.0 | 0.0 |
| dhps | chr8 | 549681 | S436A | 15.0 | 17.3 | 28.3 | 9.8 | 37.5 | 0.0 | 0.0 |
| dhps | chr8 | 549685 | G437A | 26.8 | 32.7 | 27.5 | 36.3 | 4.2 | 0.0 | 17.8 |
| dhps | chr8 | 549993 | K540E | 25.4 | 17.0 | 9.4 | 22.1 | 0.0 | 85.4 | 48.9 |
| dhps | chr8 | 550117 | A581G | 8.2 | 6.1 | 2.2 | 8.8 | 0.0 | 34.1 | 4.4 |
| k13 | chr13 | 1726431 | K189T | 14.8 | 14.9 | 18.8 | 12.3 | 54.2 | 0.0 | 6.7 |
| mdr2 | chr14 | 1956202 | I492V | 23.2 | 21.3 | 22.5 | 20.6 | 20.8 | 31.7 | 31.1 |
| mdr2 | chr14 | 1956408 | F423Y | 31.4 | 30.1 | 28.3 | 31.4 | 29.2 | 36.6 | 37.8 |

Includes all mutations that reached a prevalence >5% in any given geographic unit.

classical pattern of isolation-by-distance. This background population structure is overlaid with the clear impacts of drug resistance mutations, which cause distinct structure between East and West African parasite populations. Additionally, the use of relatively dense genome-wide SNPs allowed us to carry out relatedness analysis, revealing a handful of cases where human hosts separated by many hundreds of kilometers were infected by essentially identical clones. Given the rapid breakdown of distinct genotypes by recombination in high transmission areas, it is highly likely that these events represent relatively recent infection and migration events. With this in mind, it is interesting to note that pairwise links of high relatedness tend to fall along the Congo River, an important route of transportation in DRC. Lastly, the combination of the two MIP panels allowed us to examine extended haplotypes surrounding drug resistance genes, revealing rapid breakdown of haplotypes in the population and different signals of selection in East vs. West DRC.

We previously investigated population structure using MIPs targeting 20 microsatellites in the DRC[10], failing to detect a strong signal of population structure based upon these markers. Here we leveraged the same 552 samples as the previous study, plus additional samples from the DRC and neighboring countries, to identify clear structure with an improved SNP-based genotyping method. Our ability to detect population structure in the present study is likely due to several factors. First, the SNP panel contains nearly two orders of magnitude more markers than the previous panel. While this SNP MIP panel expanded the number of loci interrogated, we have yet to achieve the full potential of MIPs. Specifically, massively increased, multiplexed probe sets that target additional portions of the genome are feasible. MIPs have now been used in human studies to detect as many as 55,000 markers in a single reaction[22]. Second, a large number of genome-wide SNPs in this study were chosen based on high $F_{ST}$

values in publically available samples from surrounding countries. This increases our power to detect geographic differentiation, but comes at the cost of not being able to comment on the relative importance of geography vs. drug resistance, which would require random genetic sampling or alternatively whole genomes. Similarly, we should be cautious when interpreting spatial clines in population structure from our data, as we may have greater power to detect structure along some axes than others due to the unequal distribution of surrounding countries in publically available samples, although in general we have good representation in both the East–West and North–South directions.

The flexible nature of MIP panels allows for multiplex detection of SNPs associated with drug resistance in any known or putative resistance loci for which they are designed. This allowed for a more detailed evaluation of molecular markers associated with antimalarial resistance than has previously been possible in the DRC. To date, studies of antimalarial resistance markers in the DRC have focused primarily on *pfcrt* (K76T), *dhfr* (N51I, C59R, S108N, I164L), *dhps* (I431V, S436A, A437G, K540E, A581G, A613S), *pfmdr* (N86Y, F184Y, D1246Y), and a few *kelch* mutations[23–29]. The data suggests that mutations associated with artemisinin resistance remained absent in the country as of 2014. The World Health Organization identified nine mutations within the K13 propeller region that are validated in terms of their clinical phenotype of artemisinin resistance, and a further 11 mutations that are candidates associated with the phenotype of delayed clearance[30]. We identified 14 mutations within the K13 gene (Supplementary Data 2), although none of these correspond to validated or candidate artemisinin resistance mutations.

Beyond looking at mutations within drug resistance genes, differences in extended haplotypes around drug resistance genes have been used to understand evolution and spread[31]. Though not originally designed for this purpose, the genome-wide MIP

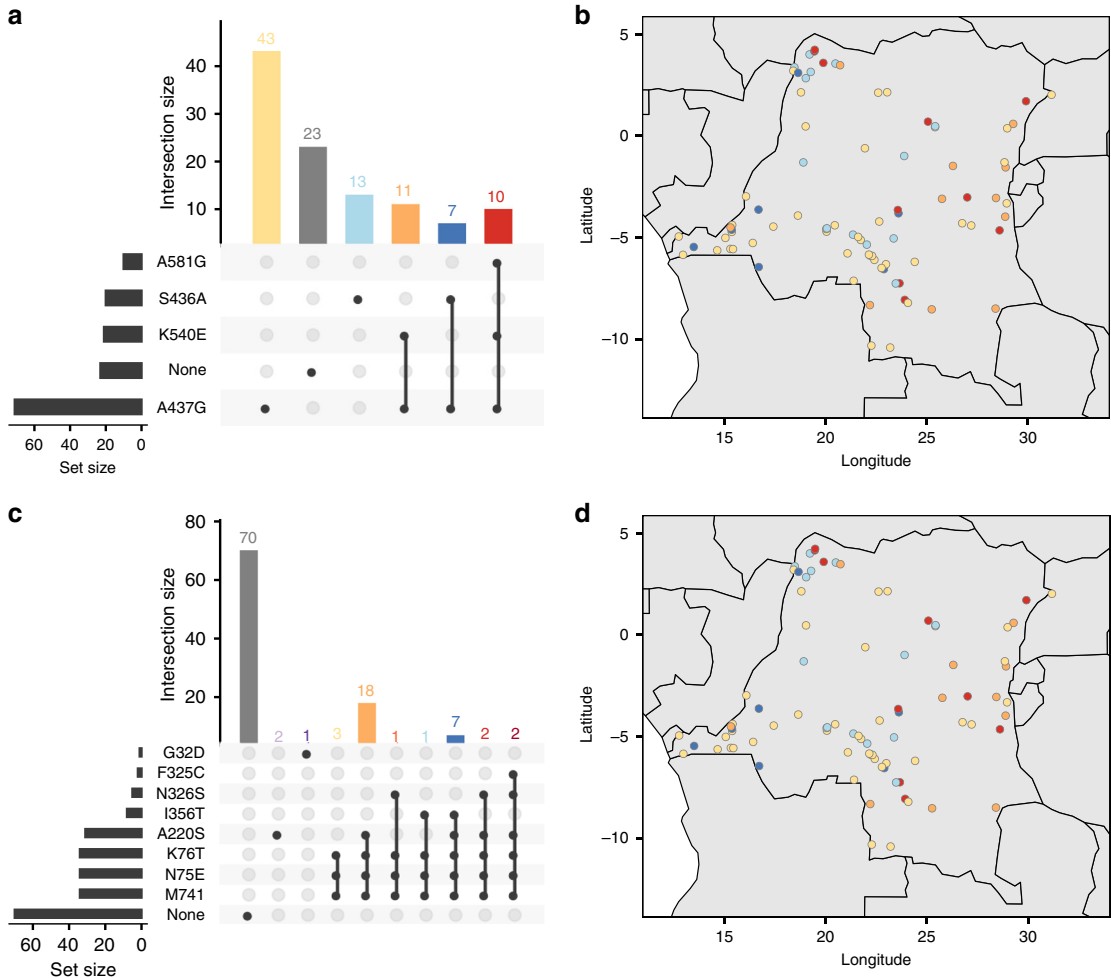

**Fig. 6 Spatial distribution of drug resistant haplotypes.** The spatial distribution of all combinations of mutant haplotypes for *dhps* and *crt* from the monoclonal DRC samples. **a**, **c** UpSetR plots showing the number of times each combination of mutations was seen for *dhps* and *crt*, respectively. **b**, **d** Show these same haplotypes on a map of DRC. Colors correspond horizontally between panels, i.e. between **a** and **b**, and between **c** and **d**, with the exception of wild-type haplotypes (gray) which are not shown in **b**, **d**.

panel can be leveraged for conducting similar analyses. For example, the differences in CV**IET** EHH between the West and East suggests that the CV**IET** haplotype in the West has potentially been more recently introduced, has experienced less breakdown through recombination, or has undergone stronger recent positive selection as compared to the East. Redesign of the selected targets with denser sampling around known drug resistance genes will allow for more robust assessment of these selected regions.

DRC's location in central Africa and the enormous number of malaria cases in the country means that malaria control in Africa likely depends on improving our understanding on Congolese malaria. This represents the largest study of falciparum population genetics in the DRC and, unlike other large population genetic studies of malaria in Africa, leverages a nationally representative sampling approach. Thus, this study provides the first data on fine-scale genetic structure of parasites at a national scale in Africa, and provides a baseline that can be used to study how implementation programs impact parasite populations in the region. The MIP platform represents a highly scalable and cost-effective means of providing genome-wide genetic data, relative to whole-genome sequencing[10]. The highly flexible nature of the platform allows it to be rapidly scaled in terms of targets and samples leading it to be applicable across malaria-endemic countries.

## Methods

**Study populations.** Chelex-extracted DNA samples from dried blood spots, collected as part of the 2013–2014 DRC Demographic Health Survey (DHS), were tested using quantitative real-time PCR to detect *Plasmodium falciparum* lactate dehydrogenase (*pfldh*)[32,33]. Previously published DRC samples[10] were included ($n = 589$), and used to set a Ct threshold of <30 for inclusion for sequencing, which was applied to the remaining DRC samples ($n = 1450$), resulting in a total of 2039 DRC samples sent for sequencing. These samples represented 369 of the overall 539 DHS clusters. In addition, dried blood spot samples from four further counties were used: Ghana ($n = 194$), Tanzania ($n = 120$), Uganda ($n = 63$), and Zambia ($n = 121$). Samples from Ghana were collected in 2014 from symptomatic RDT and/or microscopy positive individuals presenting at health care facilities in Begoro ($n = 94$) and Cape Coast ($n = 98$)[34]. Samples from Tanzania were collected in 2015 from symptomatic RDT-positive patients of all ages at Kharumwa Health Center in Northwest Tanzania[35]. Samples from Uganda were collected in 2013 from RDT-positive symptomatic patients at Kanungu in Southwest Uganda[36]. Finally, samples from Zambia were collected in 2013 from RDT-positive individuals from a community survey of all ages in Nchelenge District in northeast Zambia on the border with the DRC. All non-DRC samples were Chelex extracted, except for the Ghanaian samples which were extracted using QiaQuick per protocol (Qiagen, Hilden, Germany). This study was approved by the Internal Review Board at UNC and the Ethics Committee of the Kinshasa School of Public Health.

**Design of MIP panels.** We used two distinct MIP panels—a genome-wide panel designed to capture overall levels of differentiation and relatedness, and a drug resistance panel designed to target polymorphic sites known to be associated with antimalarial resistance (Supplementary Note 1). When selecting targets for the genome-wide panel, we used the publicly available *P. falciparum* whole-genome sequences provided by the Pf3k and *P. falciparum* Community projects from the MalariaGEN Consortium. This consisted of sample sets from Cameroon ($n = 134$),

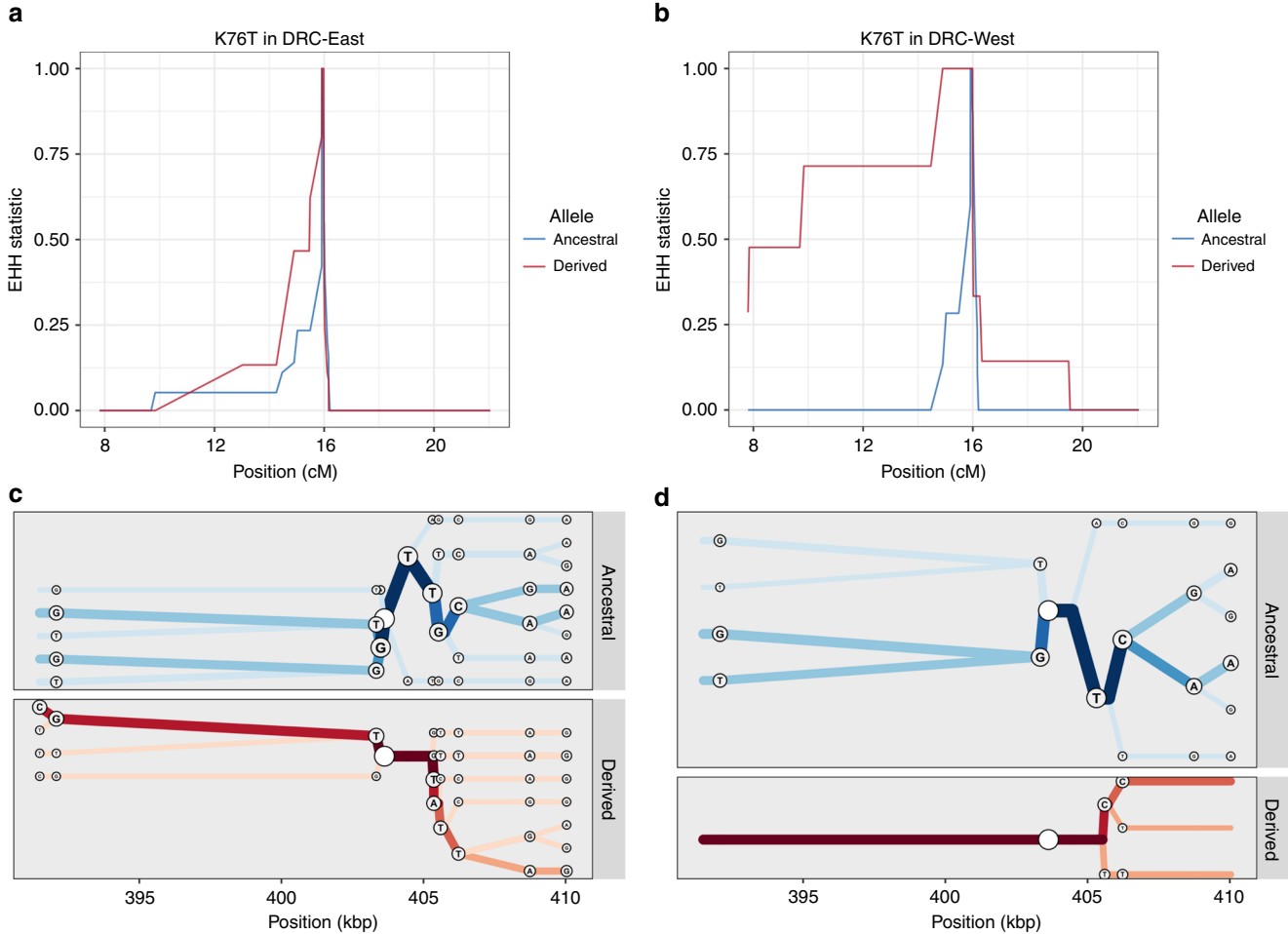

**Fig. 7 Extended haplotype homozygosity and bifurcation plots for *pfcrt* K76T. a, b** Display extended haplotype homozygosity (EHH) curves from the monoclonal samples with no missing genotype data 200 kb upstream and downstream from the K76T core single-nucleotide polymorphism in centimorgans among the samples from the eastern Democratic Republic of the Congo (DRC) and western DRC. **c, d** Show haplotype bifurcation plots with respect to the core allele ancestry and the eastern DRC and western DRC for a subsetted region. Position is considered in kilobases, and segregating sites for each haplotype are displayed at the nodes. Overall, there is strong evidence for recent positive selection of the *pfcrt* CV**IET** haplotype in the west that is mitigated in the east.

DRC ($n = 285$), Kenya ($n = 52$), Malawi ($n = 369$), Nigeria ($n = 5$), Tanzania ($n = 66$) and Uganda ($n = 12$) (Supplementary Data 1). The genomic sequence from these samples underwent alignment, variant calling, and variant-filtering following the Pf3k strategy consistent with the Genome Analysis Toolkit (GATK, version 3.6 unless otherwise indicated) Best Practices with minor modifications[37–40]. Full details of the bioinformatic pipeline used in MIP design are given in the Supplementary Note 1. Samples from Nigeria and Uganda were dropped after variant calling due to small sample sizes, and the final filtered sequences were used to calculate Weir and Cochran's $F_{ST}$[41] with respect to country for each biallelic locus. The 1000 loci with the highest $F_{ST}$ values were considered for MIP design as phylogeographically informative loci. Of these 1000 potential loci, 739 were identified as regions that were suitable for MIP-probe design. Separately, from the combined SNP file, we identified 1595 loci that had a minor-allele frequency >5%, had an $F_{ST}$ value between 0.005 and 0.2, and were annotated by SnpEff (version 4-3t) as functionally silent mutations. These loci were identified as putatively neutral SNPs, and 1151 were found to be suitable for MIP design. The distribution of MIPs is shown in Supplementary Fig. 14 and MIP sequences and targets are shown in Supplementary Data 3.

**MIP capture and sequencing**. In addition to patient samples, control samples were known mixtures of 4 strains of genomic DNA from malaria at the following ratios: 67% 3D7 (MRA-102, BEI Resources, Manasas, VA), 14% HB3 (MRA-155), 13% 7G8 (MRA-154) and 6% DD2 (MRA-156). They were also represented at two different parasite densities (29 and 467 parasites/µl). MIP capture and sequencing library preparation were carried out as described in the Supplementary Note 1[10]. Drug resistance libraries were sequenced on Illumina MiSeq instrument using 250 bp paired end sequencing with dual indexing using MiSeq Reagent Kit v2. Genome-wide libraries were sequenced on Illumina Nextseq 500 instrument using

150 bp paired end sequencing with dual indexing using Nextseq 500/550 Mid-output Kit v2.

**MIP variant calling and filtering**. MIP variant calling is summarized in the Supplementary Note 1[10]. Within each sample, variants were dropped if they had a Phred-scaled quality score of <20. Across samples, variant sites were dropped if they were observed only in one sample, or if they had a total UMI count of <5 across all samples. This data set was considered the final raw data used for additional filtering.

Additional filters were applied to both genome-wide and drug resistance datasets prior to carrying out analysis. Sites were restricted to SNPs, and in the case of the genome-wide panel these were filtered to the pre-designed biallelic target SNP sites. Any variant that was represented by a single UMI in a sample, or that had a within-sample allele frequency (WSAF), UMI count of allele/total UM less than 1%, was eliminated. Any site that was invariant across the entire dataset after this procedure was dropped. Samples were assessed for quality in terms of the proportion of low-coverage sites, where low-coverage was defined as fewer than 10 supporting UMIs. Samples with >50% low-coverage loci were dropped. Variant sites were then assessed by the same means in terms of the proportion of low-coverage samples, and sites with >50% low-coverage samples were dropped. Samples were then combined with metadata, including geographic information, and were only retained if there were at least 10 samples in a given country. This resulted in dropping Tanzanian samples from the drug resistance dataset, but no other countries were dropped. Post-filtering, genome-wide data consisted of 1382 samples (DRC = 1111, Ghana = 114, Tanzania = 30, Uganda = 45, Zambia = 82) and 1079 loci, and drug resistance data consisted of 674 samples (DRC = 557, Ghana = 29, Uganda = 43, Zambia = 45) and 1000 loci. The complete bioinformatic pipeline is shown in Supplementary Fig. 15.

**Complexity of infection**. We applied THE REAL McCOIL (v2) categorical method to the SNP genotyped samples to estimate the COI of each individual[13]. Details of the analysis are in the Supplementary Note 1.

**Analysis of population structure**. WSAFs were calculated for all genome-wide SNPs, with missing values imputed as the mean per locus. Principal component analysis (PCA) was carried out on WSAFs using the *prcomp* function in R version 3.5.1. The relative contribution of each locus was calculated from the loading values as $|l_i| / \sum_{i=1}^{L} |l_i|$, where $|l_i|$ is the absolute value of the loading at locus $i$, and $L$ is the total number of loci. PCA results were explored in a spatial context by taking the mean of the raw principal component values over all samples in a given DHS cluster, and plotting this against the geoposition of the cluster.

**Identity by descent analysis**. Pairwise IBD was calculated between all samples from the genome-wide SNPs. We used Malécot's[42] definition of $f$ as the probability of identity by descent, where $f_{uv}$ can be defined as the probability of a randomly chosen locus being IBD between samples u and v. At locus i, let $A$ denote the reference allele, which occurs at population allele frequency $p_i$, and let $a$ denote the non-reference allele, which occurs at population allele frequency $q_i = 1 - p_i$. Assuming that both samples u and v are monoclonal, let $X_{ui}$ denote the observed allele at locus i in sample u, and equivalently let $X_{vi}$ denote the observed allele in sample v. Then the probabilities of all possible observed allele combinations between the two samples can be written:

$$
\begin{aligned}
\Pr(X_{ui} = A, X_{vi} = A | f_{uv}) &= f_{uv} p_i + (1 - f_{uv}) p_i^2 \\
\Pr(X_{ui} = A, X_{vi} = a | f_{uv}) &= (1 - f_{uv}) p_i q_i \\
\Pr(X_{ui} = a, X_{vi} = A | f_{uv}) &= (1 - f_{uv}) p_i q_i \\
\Pr(X_{ui} = a, X_{vi} = a | f_{uv}) &= f_{uv} q_i + (1 - f_{uv}) q_i^2
\end{aligned}
\tag{1}
$$

from which we can calculate the likelihood of a given value of $f_{uv}$ over all loci as:

$$
L(f_{uv} | X_u, X_v) = \prod_{i=1}^{L} Pr(X_{ui}, X_{vi} | f_{uv}).
\tag{2}
$$

In practice, population allele frequencies ($p_i$) were calculated using the mean WSAF for that locus over all samples. Samples were then coerced to monoclonal by calling the dominant allele at every locus. The likelihood was evaluated using Eq. (2) in log-space for a range of values of $f_{uv}$ distributed between 0 and 1 in equal increments of 0.02. The maximum likelihood estimate $\hat{f}_{uv} = \text{argmax}_f L(f | X_u, X_v)$ was calculated between all sample pairs. Hereafter the terms IBD and $\hat{f}_{uv}$ are used interchangeably.

The validity of this method of coercing samples to monoclonal before estimating IBD via maximum likelihood was rigorously explored in a simulation-based analysis. First, a simulation framework was created that permitted simulating samples with variable polyclonality. This framework is described in detail in Supplementary Note 2. Second, true vs. estimated IBD were plotted for a range of polyclonal settings and a range of sub-sampled data sizes going down from the true data to 500, 100, and 20 SNPs. Any positive or negative bias introduced by forcing samples to be monoclonal would be reflected and quantified in this plot.

Mean IBD was calculated within and between DHS clusters, and compared using a two-sample *t*-test. Sample pairs were also binned into groups based on geographic separation (great circle distance) in 100 km bins, with an additional bin at distance 0 km to capture within-cluster comparisons. Mean and 95% confidence intervals of IBD were calculated for each group. Finally, sample pairs with IBD > 0.9 were identified, and explored in terms of their WSAFs and their spatial distribution.

**Estimating mutation prevalence from drug resistance panel**. Given previous findings of an East–West divide in molecular markers of antimalarial resistance in the DRC[8,9], all samples in the DRC were divided by geographically weighted K-means clustering into two populations. The prevalence of every mutation identified by the drug resistance MIP panel was then calculated in East and West DRC, as well as at the country level. Prevalences in each DHS cluster were used to produce smooth prevalence maps using PrevMap version 1.4.2 in R[43].

**Analysis of monoclonal haplotypes**. Results of the previous COI analysis on the genome-wide SNPs with THE REAL McCOIL were used to identify samples that were monoclonal with a high degree of confidence. Samples were defined as monoclonal if the upper 95% credible interval did not include any COI greater than one. This resulted in 408 monoclonal samples, of which 143 overlapped with the drug resistance MIP dataset and therefore could be used to explore the joint distribution of mutations in drug resistance genes. 107 of these were from DRC. Analysis focussed on the *dhps* and *crt* genes. Raw combinations of mutations were visualized using the UpSet package in R[21], and the spatial distribution of haplotypes was explored by plotting these same mutant combinations against DHS cluster geoposition.

**Extended haplotype homozygosity analysis**. In order to improve our power to detect hard-sweeps and capture patterns of linkage-disequilibrium with EHH statistics among putative drug resistance SNPs, we combined the genome-wide and

the drug resistance filtered biallelic SNPs into a single dataset (Supplementary Note 1). All associated EHH calculations were carried out using the R-package rehh (version 2.0.4), and were truncated when fewer than two haplotypes were present or the EHH statistic fell below 0.05[44,45]. In addition, we allowed EHH integration calculations to be made without respect to "borders," which were frequent due to the MIP-probe design. Although this would result in an inflated integration statistic if the EHH statistic had not yet reached 0 within the region of investigation, this problem was mitigated by only comparing between subpopulations, and not between loci. EHH decay, bifurcation plots, and haplotype plots were adapted from the rehh package objects and modified using ggplot[46].

**Reporting summary**. Further information on research design is available in the Nature Research Reporting Summary linked to this article.

## Data availability
DHS data for the 2013 DRC DHS is available here: https://dhsprogram.com/what-we-do/survey/survey-display-421.cfm. This includes clinical and GPS information and is available upon request from the DHS program. All raw sequencing data is available at the NCBI SRA (Accession numbers: PRJNA454490, PRJNA545345, and PRJNA545347).

## Code availability
Tools for MIP variant calling and filtering are available at https://github.com/bailey-lab/MIPTools (v.0.19.12.13) and https://github.com/Mrc-ide/mipanalyzer (v.1.0.0). Code and data are available for each figure at https://github.com/bobverity/antimalarial_resistance_DRC. Code access is unrestricted.

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

## Acknowledgements

This work was supported by the National Institutes of Health (R01AI107949, R01AI139520, K24AI134990, R21AI121465, F30AI143172, U19AI089680). R.V is funded by a Skills Development Fellowship: this award is jointly funded by the UK Medical Research Council (MRC) and the UK Department for International Development (DFID) under the MRC/DFID Concordat agreement and is also part of the EDCTP2 programme supported by the European Union. We would also like to thank everyone who participated in the studies and all members of the study teams in Ghana, Zambia, Uganda, and Tanzania. We would like to thank the DHS Program and USAID for the collection and access to samples from the DRC Demographic Health Survey.

## Author contributions

R.V., O.A., N.F.B., J.A.B., and J.J.J. contributed to data analysis, writing and experimental design. O.J.W. contributed to data analysis and writing. N.J.H. and A.P.M. contributed software design. M.K.M, J.B.P., P.K.T, M.C., P.J.R., D.S.I., J.N., J.G., M.M., D.E.N., W.J.M., B.A.M., J.L.M.H., A.G., and A.K.T. contributed samples from studies conducted at their sites and reviewed the paper. A.C.G. contributed to analysis design and reviewed the manuscript. P.W.M., K.T., T.F., and M.D. contributed laboratory analysis. S.R.M. contributed coordination with DRC investigators, experimental design, and writing.

## Competing interests

The authors declare no competing interests.
