## [Peer Review File · Nature Communications]

Reviewers' Comments:

Reviewer #1:

Remarks to the Author:

Verity, Aydemir, Brazeau and colleagues report on a unique malaria molecular surveillance dataset produced from genotyping 1834 SNPs in 2537 *P. falciparum* samples from the DRC using molecular inversion probes (MIPs). They evaluate the spatial distribution of genetic markers for drug resistance and examine parasite population genetic structure through analysis of relatedness. This manuscript reports both technical and biological insights. On the technical side, the authors find that this MIP panel is more sensitive to genetic structure in the parasite population than microsatellite panels or genotype data from hypervariable antigens. They show that dense genotyping data and relatedness inference are able to detect an isolation-by-distance signal of structure in the parasite population at a very fine scale (hundreds of km). Biologically, the authors show that drug resistance markers are distributed heterogeneously within the DRC, in particular markers at *pfcr* and *pfdhps*, though the latter result has been reported previously. The distribution of these markers is likely driven by heterogeneity in drug pressure, as other resistance loci (*pfhfr* and *pfmdr1*) do not show a similar pattern.

This manuscript will be of value to the community, as it clearly demonstrates the importance of high resolution molecular surveillance data for malaria. The manuscript could be improved through attention to the following issues:

1) My most serious issue deals with the approach for dealing with polyclonal infections, given the potential strong impact of polyclonality on IBD inference, PCA, etc. Given that the mean COI within the DRC is >1 , the approach of 'coercing' (line 439) samples to being monoclonal has the possibility of incorrectly phasing COI=2 samples where allelic fractions are relatively equal, and phasing a dominant strain for COI >2 samples is potentially even more fraught depending on the relative frequencies of the minor strains and the MAF of the markers. Have the authors investigated the potential analysis impacts of incorrect coercion of the dominant strain using this method? Deeper attention to this issue is necessary.

2) Line 194: Is the threshold for high relatedness used here (90% IBD) chosen for any particular reason? Would lower thresholds yield similar results?

3) Line 278: How important is the density of SNP markers for the results reported? Is it necessary to type 1834 markers, or could fewer suffice? Why not do whole genome sequencing if large numbers of markers are desirable? It could be helpful to readers to clarify the rationale behind the genotyping strategy adopted in this study.

Figure 5: Would be nice to have a scale indicator on panel b (and in maps represented in other figures) to relate to the country-level map to the distances shown in panel a. Also, is it possible to more clearly label the Congo River on the map?

Reviewer #2:

Remarks to the Author:

Verity, Aydemir, and Brazeau present a large analysis of 2,500 *Plasmodium falciparum* samples from the Democratic Republic of the Congo. Using a panel of over 1,000 SNPs genotyped using MIP capture and sequencing, the authors thoroughly characterised the local parasite population, including population structure, level of relatedness and of drug resistance. I concur with the authors that these

data set a baseline and could be very important for future surveillance in the region.

The study is overall well designed, the analysis are pretty exhaustive and extremely well detailed, and the conclusions are adequately supported. However as the use of MIP in malaria is still relatively recent, some clarification might be needed to strengthen the results. I report some suggestion below which, while I believe would benefit the readers, I do not expect will significantly impact any of the high level conclusions.

MIP panel

- Unless I misunderstood, it appears that the authors are genotyping known variable positions in the genome based on external data. If that is the case, I don't understand while they proceed with a further variant discovery and filtration step (supplementary methods, pp. 3-4), rather than directly genotyping those positions. In fact, I'm not confident the approach used is necessarily appropriate. Namely, WGS data from genetic crosses are used for calibrating the variant filtration step (i.e. the process where a genome position is analysed to decide whether it is truly variable). However, due to the profound technological differences between WGS and MIP, I'm not sure that the former constitutes a representative "gold standard" for the latter as coverage profiles, mapping process, etc are very different. At very least, authors should provide a sense of how many variants are discovered and filtered out out of those targeted (I'd expect this number to be very little, given the authors are typing known variable positions). I do agree however that even true variants are not guaranteed to be accessible and typable using MIPs, but I believe this aspect is taken care of by the subsequent steps.
- After defining the set of variants, the next question is the genotyping accuracy. Specifically, the authors show that using a set of artificial mixtures they can achieve extremely high correlation between expected and estimated within-samples allele frequencies (supplementary figure 2). While the trend is very reassuring, I'm not sure r^2 is the best metric to be reported here given the large number of ties (the expected allele frequencies form a discreet and very limited set of values). Moreover, more than half of the sites are invariant, and that would artificially inflate the correlation coefficient. At very least those value should be excluded in the calculation. Ideally, other measure should be also reported, e.g. relative error rate.
- As a minor note, the authors use a panel of 1,151 SNPs deemed to be putatively neutral. However the selection criteria impose those variant to be highly variable and highly differentiated, which is usually a signature of balancing selection. While the authors also require the SNPs to be 'silent', linkage disequilibrium could still be playing a role.

Identity by descent

- It would be extremely valuable to provide some reassurance that complex infections are not significantly impacting the results. I accept the authors imposed a ploidy of 1 in the genotyping process to make the problem tractable. However, as they report, the vast majority of the samples have a COI greater than 2 so the above could potentially represent a gross approximation. Ideally, it would be useful to evaluate and quantify the impact (e.g. via simulations or using the artificial mixtures), but at very least results should be interpreted in a very cautionary way.
- In the light of the point above, I would take carefully the statements re the near-identical distant samples. In fact, from supplementary figure 6, it appears that a good proportion of those allegedly identical pairs are actually quite different. Here I considered the 0.5 threshold as an illustrative yet arbitrary way to quantify discordances, unless there there are analytical reasons for that (e.g. if derived from the within-sample allele frequency accuracy)? What is the average number of matches and mismatches between any given pair? Are those numbers really unusually high (as I'd expected, if IBD is function of genetic distance)?
- Incidentally, from the same figure, I find the number of hom alt calls relative to the number of hom ref calls to be unusually high (~700 vs 300). Is this a result of the MIP panel which is implicitly enriching for high diversity? What is the average number of hom ref and hom alt calls across the

dataset?

- As a minor note, I find the definition of IBD provided (ll. 181-183) puzzling. While it's certainly true in the spirit, I'm not sure I would characterise IBD as a better measure than IBS because it takes allele frequencies into account. Identity-by-state refers to a static view of a locus (i.e. being the same), while identity-by-descent captures its evolutionary process (i.e. having the same, usually recent, ancestry). Two allele can be identical by state and not by descent.

Roberto Amato, Wellcome Sanger Institute

The Impact of Antimalarial Resistance on the Genetic Structure of *Plasmodium falciparum* in the DRC: Response to reviewers' comments

Reviewer #1 (Remarks to the Author):

Verity, Aydemir, Brazeau and colleagues report on a unique malaria molecular surveillance dataset produced from genotyping 1834 SNPs in 2537 *P. falciparum* samples from the DRC using molecular inversion probes (MIPs). They evaluate the spatial distribution of genetic markers for drug resistance and examine parasite population genetic structure through analysis of relatedness. This manuscript reports both technical and biological insights. On the technical side, the authors find that this MIP panel is more sensitive to genetic structure in the parasite population than microsatellite panels or genotype data from hypervariable antigens. They show that dense genotyping data and relatedness inference are able to detect an isolation-by-distance signal of structure in the parasite population at a very fine scale (hundreds of km). Biologically, the authors show that drug resistance markers are distributed heterogeneously within the DRC, in particular markers at *pfcr* and *pfdhps*, though the latter result has been reported previously. The distribution of these markers is likely driven by heterogeneity in drug pressure, as other resistance loci (*pfdhfr* and *pfmdr1*) do not show a similar pattern.

This manuscript will be of value to the community, as it clearly demonstrates the importance of high resolution molecular surveillance data for malaria. The manuscript could be improved through attention to the following issues:

1) My most serious issue deals with the approach for dealing with polyclonal infections, given the potential strong impact of polyclonality on IBD inference, PCA, etc. Given that the mean COI within the DRC is >1 , the approach of 'coercing' (line 439) samples to being monoclonal has the possibility of incorrectly phasing COI=2 samples where allelic fractions are relatively equal, and phasing a dominant strain for COI >2 samples is potentially even more fraught depending on the relative frequencies of the minor strains and the MAF of the markers. Have the authors investigated the potential analysis impacts of incorrect coercion of the dominant strain using this method? Deeper attention to this issue is necessary.

Response: We thank the reviewer for this comment. We have now undertaken a comprehensive simulation-based analysis to explore the potential impact on IBD inference of coercing polyclonal samples to monoclonal. First, we have formulated a structured simulation model that can generate genetic datasets under a range of polyclonality, and can capture different degrees of superinfection and cotransmission. This simulation model is described in detail in Supplemental Text 2. The output of this model includes both "true" haplotypes that are phased and contain zero sequencing errors, and "observed" genotypes that are obtained by simulating strain frequencies and read counts before coercing to monoclonal. We then used this framework to simulate true IBD under a range of settings, including low and high polyclonality, and we evaluated the maximum likelihood estimator on "observed" genotypes for the same simulations. Supplemental Figure 6 shows the comparison between true simulated IBD and that estimated by our maximum likelihood method. As we would expect, in monoclonal settings we have very high accuracy, although in highly polyclonal settings we

underestimate IBD due to many IBD relationships occurring between minor strains. Hence, we can say that our estimator is under-powered, but is conservatively biased. This gives us confidence that our IBD-based results and conclusions are in fact robust, as if anything we are missing certain highly related sample pairs.

The main paper now contains the added text (Page 8, lines 173-180):

“We first carried out a simulation-based analysis to explore the accuracy of our maximum likelihood estimator (see Supplemental Text 2 and Supplemental Figure 6), finding that we were conservatively biased in cases of high polyclonality. Hence, we expect to underestimate true IBD by this method. This result did depend on the number of genotyped positions, with estimates becoming increasingly unreliable for smaller datasets of 100 or 20 SNP loci. In the real data, the overall distribution of pairwise IBD was found to be heavy-tailed, consisting of a large body of weakly related samples and a tail of very highly related samples”

The methods section also contains the added text (Page 18, lines 413-419):

“The validity of this method of coercing samples to monoclonal before estimating IBD via maximum likelihood was rigorously explored in a simulation-based analysis. First, a simulation framework was created that permitted simulating samples with variable polyclonality. This framework is described in detail in Supplemental Text 2. Second, true vs. estimated IBD were plotted for a range of polyclonal settings and a range of sub-sampled data sizes going down from the true data to 500, 100 and 20SNPs. Any positive or negative bias introduced by forcing samples to be monoclonal would be reflected and quantified in this plot.”

We note that earlier results in the paper, such as the PCA, make use of complete within-sample allele frequencies and do not require coercing to monoclonal, hence we believe these results to be justifiable even given the highly polyclonal setting.

2) Line 194: Is the threshold for high relatedness used here (90% IBD) chosen for any particular reason? Would lower thresholds yield similar results?

Response: The threshold of 90% IBD was selected as a lower limit of putative clonal transmission events, where each sample could contribute at most 5% genotyping error to the pairwise comparison. Exploration of these events is of interest, as in high transmission settings they represent likely transmission in the preceding generation, or a single mosquito inoculating different hosts. The latter case is unlikely given the geographic distance between pairs with high relatedness, as shown by Figure 5. Further exploration of DRC pairs that are at least meiotic siblings (n=62) is a separate project that we hope to publish in the near future.

3) Line 278: How important is the density of SNP markers for the results reported? Is it necessary to type 1834 markers, or could fewer suffice? Why not do whole genome sequencing if large numbers of markers are

desirable? It could be helpful to readers to clarify the rationale behind the genotyping strategy adopted in this study.

Response: In terms of comparison to whole genome sequencing, the main barrier here is cost. There would certainly be as much or more information in whole genomes, but this would be orders of magnitude more expensive than MIPs for the sampling design carried out here. It would also be difficult to obtain sufficient parasitaemias for this to be feasible.

In terms of comparison to smaller numbers of loci, we have now explored this systematically as part of the simulation-based analysis described above. We find that we would have obtained similar results for our IBD analysis using around 500 markers, rather than the full 1079, but smaller numbers (e.g. 100, 20 markers) begins to introduce significant positive bias into our IBD estimates. Changes to the main text are described in the response to reviewer 1 comment 1.

Figure 5: Would be nice to have a scale indicator on panel b (and in maps represented in other figures) to relate to the country-level map to the distances shown in panel a. Also, is it possible to more clearly label the Congo River on the map?

Response: We have now added scale indicators to the edges in Figure 5, and a label pointing to the Congo river. The new figure legend reads (changed text in red) (Page 20, lines 476-479):

“Panel (a) shows the mean IBD between clusters, binned by the spatial distance between clusters. Vertical lines show 95% confidence intervals. Panel (b) shows the spatial distribution of highly related (IBD>0.9) parasite pairs. **Values above edges give distances in km.** Black areas indicate major water bodies, including the Congo River **which is labelled.**”

Reviewer #2 (Remarks to the Author):

Verity, Aydemir, and Brazeau present a large analysis of 2,500 Plasmodium falciparum samples from the Democratic Republic of the Congo. Using a panel of over 1,000 SNPs genotyped using MIP capture and sequencing, the authors thoroughly characterised the local parasite population, including population structure, level of relatedness and of drug resistance. I concur with the authors that these data set a baseline and could be very important for future surveillance in the region.

The study is overall well designed, the analysis are pretty exhaustive and extremely well detailed, and the conclusions are adequately supported. However as the use of MIP in malaria is still relatively recent, some clarification might be needed to strengthen the results. I report some suggestion below which, while I believe would benefit the readers, I do not expect will significantly impact any of the high level conclusions.

MIP panel

- Unless I misunderstood, it appears that the authors are genotyping known variable positions in the genome based on external data. If that is the case, I don't understand while they proceed with a further variant discovery and filtration step (supplementary methods, pp. 3-4), rather than directly genotyping those positions. In fact, I'm not confident the approach used is necessarily appropriate. Namely, WGS data from genetic crosses are used for calibrating the variant filtration step (i.e. the process where a genome position is analysed to decide whether it is truly variable). However, due to the profound technological differences between WGS and MIP, I'm not sure that the former constitutes a representative "gold standard" for the latter as coverage profiles, mapping process, etc are very different. At very least, authors should provide a sense of how many variants are discovered and filtered out out of those targeted (I'd expect this number to be very little, given the authors are typing known variable positions). I do agree however that even true variants are not guaranteed to be accessible and typable using MIPs, but I believe this aspect is taken care of by the subsequent steps.

Response: The reviewer is correct that we are genotyping our samples for loci known to be variable from external data, i.e. the MIP panel was designed on the basis of variation in Pf3K. Given our raw MIP data at these known loci, we then perform filtering but not further variant discovery. This filtering step was necessary due to poor coverage in certain samples and loci, which added significant noise but little signal to the overall data.

Following the suggestion to report the number of variants filtered at each stage, we have added a flowchart (Supplemental Figure 15) to better describe the complete MIP panel design and post-processing pipeline. We have also signposted this figure in the methods section:

- After defining the set of variants, the next question is the genotyping accuracy. Specifically, the authors show that using a set of artificial mixtures they can achieve extremely high correlation between expected and estimated within-samples allele frequencies (supplementary figure 2). While the trend is very reassuring, I'm not sure r^2 is the best metric to be reported here given the large number of ties (the expected allele frequencies form a discreet and very limited set of values). Moreover, more than half of the sites are invariant, and that would artificially inflate the correlation coefficient. At very least those value should be excluded in the calculation. Ideally, other measure should be also reported, e.g. relative error rate.

Response: Although it is valuable to show the correlation between the expected and observed values for the invariant sites, the reviewer is correct in that this will inflate the R^2 value. We removed the invariant sites from the correlation coefficient calculations as the reviewer suggests. In addition, we added the average relative error values as a new panel to the Supplemental Figure 2. The new figure legend now reads as follows (changed text in red):

"A mix of 4 laboratory strains (as described in the methods) were used as controls. (a) Each targeted SNP's expected allele frequency (in increasing order) is plotted in blue based on which strains harbor the SNP and what ratio the strain was mixed in the sample. Each SNP's frequency as measured experimentally is plotted in red. Pearson's correlation coefficient between the expected and observed frequencies, **calculated for alleles SNPs with nonzero expected**

frequencies, were $R=0.925968$ ($R^2=0.856938$). (b) Average relative error was also calculated for each SNP using the formula $(\text{experimental mean frequency} - \text{expected frequency}) / \text{expected frequency}$. These values were grouped according to the expected frequencies and average for each group were plotted (error bars showing 95% CI). In total, 114 control reactions were run with experiments."

- As a minor note, the authors use a panel of 1,151 SNPs deemed to be putatively neutral. However the selection criteria impose those variant to be highly variable and highly differentiated, which is usually a signature of balancing selection. While the authors also require the SNPs to be 'silent', linkage disequilibrium could still be playing a role.

Response: We point to the following sentence in reference to the putative neutral loci from the Supplemental Text:

"Separately, from the combined SNP file, we identified 1,595 potential loci that had a minor-allele frequency greater than 5%, had an F_{ST} value between 0.005 and 0.2, and were annotated by SNPEff as functionally silent mutations. These were identified as putatively neutral SNPs. Of these 1,595 potential loci, 1151 were suitable for MIP-probe design. 76 loci were shared between phylogeographically informative and putatively neutral loci."

Although we did impose F_{ST} bounds on the putatively neutral alleles, these should have removed rare allele or alleles at fixation but not necessarily selected for highly differentiated variants. This is in contrast to the putative geographically informative loci, which were chosen on the basis of high F_{ST} . Hence we do agree that among the geographically informative variants, some of the variants may be undergoing balancing selection (as can be seen by the divide among the East and West DRC allele frequency clines and signatures of selection).

Identity by descent

- It would be extremely valuable to provide some reassurance that complex infections are not significantly impacting the results. I accept the authors imposed a ploidy of 1 in the genotyping process to make the problem tractable. However, as they report, the vast majority of the samples have a COI greater than 2 so the above could potentially represent a gross approximation. Ideally, it would be useful to evaluate and quantify the impact (e.g. via simulations or using the artificial mixtures), but at very least results should be interpreted in a very cautionary way.

Response: See response to reviewer 1 comment 1.

- In the light of the point above, I would take carefully the statements re the near-identical distant samples. In fact, from supplementary figure 6, it appears that a good proportion of those allegedly identical pairs are actually quite different. Here I considered the 0.5 threshold as an illustrative yet arbitrary way to quantify discordances, unless there are analytical reasons for that (e.g. if derived from the within-sample allele

frequency accuracy)? What is the average number of matches and mismatches between any given pair? Are those numbers really unusually high (as I'd expected, if IBD is function of genetic distance)?

Response: Yes, the 0.5 threshold is intended as an illustrative way to quantify discordances. We chose not to conduct a more sophisticated identity-in-state-based analysis using within-sample allele frequencies because we argue instead for the use of IBD over IBS where possible.

The average proportion of matches between any given pair is 74.5%, with the 95% quantiles of the distribution spanning 70.0%-80.3%. In contrast, the smallest observed proportion in the 12 identified highly related sample pairs is 95.2% (sample 178 vs. 909), which puts these sample pairs in the bottom 0.0037% of the distribution. So these samples are significant outliers in terms of the number of matches and mismatches.

- Incidentally, from the same figure, I find the number of hom alt calls relative to the number of hom ref calls to be unusually high (~700 vs 300). Is this a result of the MIP panel which is implicitly enriching for high diversity? What is the average number of hom ref and hom alt calls across the dataset?

Response: Admittedly, this figure was confusing as axes labels are referent within-sample allele frequency. We have updated the figure such that axis labels now read "REF-WSAF". As Dr. Amato points out, the number of homo ref calls is much higher than homo-alt calls as would be expected. The figure legend now reads (changed text in red):

"For the 12 sample pairs identified as highly related (IBD>0.9), scatterplots compare the raw within-sample allele frequencies (WSAF) of the referent allele at every locus"

- As a minor note, I find the definition of IBD provided (ll. 181-183) puzzling. While it's certainly true in the spirit, I'm not sure I would characterise IBD as a better measure than IBS because it takes allele frequencies into account. Identity-by-state refers to a static view of a locus (i.e. being the same), while identity-by-descent captures its evolutionary process (i.e. having the same, usually recent, ancestry). Two allele can be identical by state and not by descent.

Response: This point is well taken. The text now reads (changed text in red) (Page 8, lines 169-173):

The relatedness of all pairs of samples was explored through pairwise identity by descent (IBD), estimated using a maximum likelihood approach. IBD describes the relatedness of samples in terms of their shared evolutionary history, and consequently is not influenced by a particular allele frequency distribution. This makes it a better measure than simple identity by state (IBS) when comparing between studies, as values can be compared directly.

Roberto Amato, Wellcome Sanger Institute

Reviewers' Comments:

Reviewer #1:

Remarks to the Author:

I am satisfied with the authors' response to my comments and those of the other reviewer, and support publication of this manuscript.

Reviewer #2:

Remarks to the Author:

Overall I'm happy that the Verity, Aydemir, Brazeau, and colleagues have addressed to satisfaction all the points raised. I have some further comments specifically on the novel work that has been added in response to our comments, but they are minor and I'm not requesting to address them necessarily.

- I'm satisfied with the clarifications on how variants were filtered and with the additional analysis on genotyping accuracy.

- On the putatively neutral loci, I still believe that given the excess of rare variants in African parasites, imposing a frequency cutoff of 5% and requiring a relatively high F_{ST} (up to 0.2), will enrich for non-neutral loci. However I only have circumstantial evidence to substantiate this point (hence it was a minor note to begin with) so I'm happy to accept the authors' rebuttal.

- I praise the extensive work carried out to analyse the impact of coerced polyclonal infections on IBD estimates, I think it adds a lot of robustness to the paper. At a high level, I do agree that the coercion doesn't seem to dramatically bias the results and I'm happy to accept, at a qualitative level, the authors' conclusions. I have to admit, however, that on a quantitative level things are not clear cut. My interpretation of Supp Figure 6 is that depending on the combination of parameters the authors' approach tends to either overestimate or underestimate IBD. But I do agree that underestimation seems to be much more severe than overestimation (in particular with a large SNPs panel) and thus I agree with their generalization. On a side note, the average population IBD is not necessarily the most informative metric for validation (as I expect to be ~ 0 in most population). Given the focus of the paper on highly related pairs, perhaps relative error or a scatter (density) plot of estimated IBD value in any given pair vs their true value would have been more accurate ways to characterise the bias. I understand that given the number of free parameters in the model it's not feasible to produce those plots in an intelligible way but perhaps a subset of illustrative scenarios might be enough (e.g. fixing the number of SNPs and trying either low m/λ vs high m/λ for a couple of N values?). While I think it would be useful, admittedly it won't add much to the authors' conclusions so I'm happy to leave it to their discretion (or to a different paper altogether).

- I accept that pairs are significant outliers in terms of the number of matches. I still find puzzling that they can be up to 5% different, but perhaps that's due to genotyping error (which seems to be quite high if so)?

Reviewer #1 (Remarks to the Author):

I am satisfied with the authors' response to my comments and those of the other reviewer, and support publication of this manuscript.

Thank you.

Reviewer #2 (Remarks to the Author):

Overall I'm happy that the Verity, Aydemir, Brazeau, and colleagues have addressed to satisfaction all the points raised. I have some further comments specifically on the novel work that has been added in response to our comments, but they are minor and I'm not requesting to address them necessarily.

Thank you.

- I'm satisfied with the clarifications on how variants were filtered and with the additional analysis on genotyping accuracy.

Thank you.

- On the putatively neutral loci, I still believe that given the excess of rare variants in African parasites, imposing a frequency cutoff of 5% and requiring a relatively high F_{ST} (up to 0.2), will enrich for non-neutral loci. However I only have circumstantial evidence to substantiate this point (hence it was a minor note to begin with) so I'm happy to accept the authors' rebuttal.

Although we acknowledge this point, we selected our “putatively neutral” SNPs to be sites that were identified as functionally silent mutations. As a result, fixations within populations may be the result of genetic drift, which would result in a high F_{st} , and not necessarily selection. Moreover, if these neutral loci with high F_{st} values are a result of genetic drift, these loci are ancient (given the large effective population size of African *falciparum* parasites) and are unlikely to be targets of selection on standing variation (given their effect is predicted to be “silent”).

- I praise the extensive work carried out to analyse the impact of coerced polyclonal infections on IBD estimates, I think it adds a lot of robustness to the paper. At a high level, I do agree that the coercion doesn't seem to dramatically bias the results and I'm happy to accept, at a qualitative level, the authors' conclusions. I have to admit, however, that on a quantitative level things are not clear cut. My interpretation of Supp Figure 6 is that depending on the combination of parameters the authors' approach tends to either overestimate or underestimate IBD. But I do agree that underestimation seems to be much more severe than overestimation (in particular with a large SNPs panel) and thus I agree with their generalization. On a side note, the average population IBD is not necessarily the most informative metric for validation (as I expect to be ~ 0 in most population). Given the focus of the paper on highly related pairs, perhaps relative error or a scatter (density) plot of estimated IBD value in any given pair vs their true value would have been more accurate ways to characterise the bias. I understand that given the number of free parameters in the model it's not feasible to produce those plots in an intelligible way but perhaps a subset of illustrative scenarios might be enough (e.g. fixing the number of SNPs and trying

either low m/λ vs high m/λ for a couple of N values?). While I think it would be useful, admittedly it won't add much to the authors' conclusions so I'm happy to leave it to their discretion (or to a different paper altogether).

We agree that there are some interesting patterns here that warrant further investigation. For example, the sensitivity analysis shows that for a very small number of loci (e.g. 20) the estimator appears to have an upward bias in IBD estimation. This is not unexpected for a maximum-likelihood estimator, and hopefully should direct readers towards appropriate choice of statistical methods and numbers of loci. For our purposes we believe this analysis shows that for the actual number of SNPs used in our analysis (1079) there is very little upward bias, and if anything IBD is underestimated. We are aware that designing statistical approaches for inferring IBD in the presence of polyclonal infections is an active area of research, and we hope that in future we will have better methods at our disposal that lead to more efficient use of data.

- I accept that pairs are significant outliers in terms of the number of matches. I still find puzzling that they can be up to 5% different, but perhaps that's due to genotyping error (which seems to be quite high if so)?

Genotyping error will certainly play a role here, but we also cannot rule out the effects of mixed infections. For some pairwise comparisons (e.g. sample 493 vs. 701) it is clear from the scatterplot that at least one sample is likely polyclonal, with a low-density strain leading to a spread in within-sample allele frequencies (WSAF). Hence we are looking for IBD between dominant strains of mixed infections, which will only be possible when the major strain makes up a large proportion of the WSAF. Our cutoff of $IBD > 0.9$ therefore makes an implicit cutoff in terms of the number of differences between samples. This is the best we can do in the absence of IBD methods that account for polyclonality, or sufficient coverage to phase samples.